# Differential Effects of MS Therapeutics on B Cells—Implications for Their Use and Failure in AQP4-Positive NMOSD Patients

**DOI:** 10.3390/ijms21145021

**Published:** 2020-07-16

**Authors:** Jan Traub, Silke Häusser-Kinzel, Martin S. Weber

**Affiliations:** 1Department of Neurology, University Medical Center, 37075 Göttingen, Germany; traub.jan@gmail.com; 2Institute for Neuropathology, University Medical Center, 37075 Göttingen, Germany; silke.kinzel@med.uni-goettingen.de

**Keywords:** neuromyelitis optica spectrum disorders, multiple sclerosis, B cells, dimethyl fumarate, glatiramer acetate, fingolimod, natalizumab, ocrelizumab, interferon-β, alemtuzumab

## Abstract

B cells are considered major contributors to multiple sclerosis (MS) pathophysiology. While lately approved disease-modifying drugs like ocrelizumab deplete B cells directly, most MS medications were not primarily designed to target B cells. Here, we review the current understanding how approved MS medications affect peripheral B lymphocytes in humans. These highly contrasting effects are of substantial importance when considering these drugs as therapy for neuromyelitis optica spectrum disorders (NMOSD), a frequent differential diagnosis to MS, which is considered being a primarily B cell- and antibody-driven diseases. Data indicates that MS medications, which deplete B cells or induce an anti-inflammatory phenotype of the remaining ones, were effective and safe in aquaporin-4 antibody positive NMOSD. In contrast, drugs such as natalizumab and interferon-β, which lead to activation and accumulation of B cells in the peripheral blood, lack efficacy or even induce catastrophic disease activity in NMOSD. Hence, we conclude that the differential effect of MS drugs on B cells is one potential parameter determining the therapeutic efficacy or failure in antibody-dependent diseases like seropositive NMOSD.

## 1. Introduction

Within the last decade, growing evidence has supported the involvement of B lymphocytes in the pathophysiology of neuroinflammatory diseases like multiple sclerosis (MS) and neuromyelitis optica spectrum disorders (NMOSD) [1]. The astounding success of B cell-depleting medications not only in MS but also in NMOSD was the most compelling proof for the contribution of B cells to disease pathology [2,3]. In both conditions, it is believed that cellular B cell functions like antigen-presentation, leading to activation of autoreactive T cells, and cytokine production majorly contribute to disease progression [4,5]. Regarding antibody production, there are relevant differences between MS and NMOSD: While the discovery of oligoclonal bands in the cerebrospinal fluid of MS patients dates back 60 years, it is still unclear whether these intrathecally produced immunoglobulins are of pathogenic relevance [6,7]. Originating from the central nervous system (CNS) resident plasma cells, antibodies were found to co-localize, complementing areas of ongoing CNS demyelination without affecting astrocytes [8,9]. Thus, intrathecal immunoglobulin G (IgG) synthesis is a key biological feature of MS [10]. Antigens like contactin-2, neurofascin and myelin basic protein have been suggested as potential targets, but none of them could be confirmed [11,12,13].

In contrast, autoantibodies directly targeting antigens within the CNS produced by peripheral B cells with the support of T follicular helper cells are key elements in the current pathophysiological understanding of NMOSD [14]. In seropositive patients, which represent 70% of all NMOSD patients, these autoantibodies were shown to target aquaporin-4 (AQP4), the most abundant water channel in the CNS, located in the astrocytic processes at the blood–brain barrier leading to a direct destruction of astrocytes [15,16]. Cortical demyelination occurs only at late stages of the disease as a consequence of irreversible astrocyte loss [17]. While the number of anti-AQP4-antibody-producing plasmablasts is strongly elevated in the peripheral blood and peaks at relapses [18], they are only infrequently found in the cerebrospinal fluid (CSF) of NMOSD patients [19]. This suggests that anti-AQP4 antibodies are mainly produced in the periphery and NMOSD may be considered a peripheral humoral autoimmune disorder [14]. Conclusively, oligoclonal bands are only found in 15–30% of NMOSD patients and mostly disappear during disease progression [20]. It is thought that anti-AQP4 antibodies require an inflammation-induced opening of the blood-brain barrier (BBB)in order to penetrate the CNS. Prior to NMOSD relapses, a subset of patients showed signs of viral infections [21]. In this line, disease progression in NMOSD only occurs in acute relapses, when immune cells enter the CNS tissue [22]. However, recent reports show that IgG antibodies themselves may increase the permeability of the BBB [23]. In addition, T cells are thought to be required for the disruption of the blood–brain barrier and the orchestration of the immune attack in the CNS [24]. Compared to MS, NMOSD patients display elevated interleukin-6 and interleukin-17 levels in the CSF along with a higher proportion of circulating T helper 17 and interleukin-17-producing cytotoxic T cells [24,25].

Clinical core symptoms of NMOSD include optic neuritis, acute myelitis, area postrema syndrome (hiccups, nausea and vomiting), acute brainstem syndrome, symptomatic narcolepsy and acute diencephalic clinical syndrome [26]. For anti-AQP4-IgG seropositive patients, the presence of at least one core manifestation suffices for NMOSD diagnosis, while seronegative patients require two manifestations including optic neuritis, myelitis or area postrema syndrome [26]. In the past, the term NMOSD also included a subset of AQP4-negative patients with an NMOSD phenotype and anti-myelin oligodendrocyte glycoprotein (MOG) antibodies [27]. MOG is expressed on surfaces of oligodendrocytes and myelin sheaths [28]. This condition is now considered a separate disease entity, immunopathogenetically distinct from both MS and NMOSD and called ‘MOG-IgG-associated encephalomyelitis’ [29].

As analogue clinical features and an often-relapsing disease course are seen both in MS and NMOSD, the correct diagnosis is often hard to find [28]. Especially the fact that about 20% of NMOSD patients are anti-AQP4-seronegative often complicates diagnostic procedures [30]. The recent development of numerous MS medications with diverse modes of actions has led to experimental or unintended usage of MS medications in NMOSD. For some drugs, this has caused delirious exacerbations and relapses in NMOSD. In this regard, it stands to reason that in B cell-driven diseases like NMOSD, drugs downregulating B cell inflammatory conditions may be effective while those fostering B cell inflammation may be harmful. In contrast, the role of peripheral B cell activation in the pathophysiology of MS appears more complex and may differ interindividually as intrathecal B cell function contributes to myelin loss. Thus, in this review, we attempt to address the question, how currently approved MS therapeutics alter peripheral B cell function and how these effects may contribute to their use or failure in AQP4-positive NMOSD.

## 2. Effects of MS Medications on Peripheral B Cells in Humans

As of 2020, eleven (United States: twelve) different disease-modifying therapies are approved for the treatment of relapsing MS, while azathioprine and cyclophosphamide may be considered in severe cases and daclizumab was withdrawn from the market in 2018 [31]. According to their mechanism of action, one may broadly distinguish inhibitors of proliferation (2.1), inhibitors of migration (2.2), depleting agents (2.3) and immunomodulatory drugs (2.4; Table 1). As B cells have retrieved increasing attention over the last years, B cell-directed effects of MS medications, that were primarily not developed to influence B cells, have been elucidated more and more. Within all included studies, total B cell counts were found by gating a cluster of differentiation (CD)19^+^ cells in the lymphocytic compartment. While the detailed B cell subset gating differs from study to study, there is a common sense that regulatory B cells (highly enriched within the transitional B cell subset) express high levels of CD24 and CD38. Those cells are thought to perform regulatory functions by producing interleukin-10 and thereby suppressing T cell activity [32]. In contrast, antigen-activated and memory B cells express high levels of CD27. This molecule is most often used to define these proinflammatory, interleukin-6- and TNF-secreting B cells. Memory B cells were shown to majorly contribute to disease activity in MS [33]. Circulating plasmablasts are thought to produce antibodies and drive NMOSD pathology, while they also exert regulatory functions. Moreover, some investigations also evaluated the effect of a MS medication on cytokine production and antigen-presentation of B lymphocytes. Table 1 summarizes the following findings including effects of MS therapeutics on immunoglobulin, interleukin-6 and B cell activating factor (BAFF) levels in the patients’ sera: BAFF, also known as ‘B lymphocyte stimulator’ and produced by various cell types including monocytes, dendritic cells and bone marrow stromal cells, can be cleaved from its transmembrane form, generating a soluble protein fragment. When binding the BAFF-receptor on B cells, it activates classical and noncanonical NF-κB signaling pathways. Soluble BAFF levels inversely correlate with peripheral B cell numbers and the expression of BAFF receptors. [34]. High serum BAFF levels in turn are associated with rising antibody levels in various autoimmune diseases [35].

### 2.1. Inhibition of Proliferation

Cladribine is an orally taken purine nucleoside analogue selectively depleting peripheral lymphocytes without a major impact on the innate immune system [36]. It has a greater impact on B cells than T cells, resulting in strongly decreased absolute B cell counts [4,37]. A recent investigation found a reduced memory B cell count and a lower expression of the activation marker CD25 on B cells in cladribine treated patients [38,39]. In contrast, the immature B cell count was elevated and the plasmablast count remained unaltered [38]. Cladribine exposure induces a sustained anti-inflammatory shift in the cytokine profile of surviving peripheral blood mononuclear cells with an elevated interleukin-10 and an unchanged interleukin-6 level [40]. Cladribine is also used in Morbus Waldenström, where patients benefit from a reduction of serum IgM levels [41]. To date, there are no reports about the effect of cladribine on BAFF levels.

Teriflunomide reversibly inhibits dihydro-orotate dehydrogenase, leading to a reduction in proliferation of activated T and B lymphocytes without causing cell death [42]. It is taken daily orally and several studies reported a reduction of the overall lymphocyte and B cell count. Within the decreased B cell pool, no major frequency shifts regarding B cell subsets could be observed [43,44]. Recently, a decrease in the percentage of plasmablasts after 6 months of treatment has been reported [44]. MS patients treated with teriflunomide experienced a decrease in Epstein Barr virus-specific IgG titers after 12 months of treatment [45]. To date, no data is available regarding the effect of teriflunomide on BAFF levels.

Mitoxantrone is a type II topoisomerase inhibitor, which disrupts deoxyribonucleic acid synthesis and repair [46]. Thus, after intravenous application, it inhibits lymphocyte proliferation, leading to a strong decrease of absolute total B cell lymphocyte counts [47,48,49]. It has also been shown that the proportion of memory B cells was decreased upon treatment. Moreover, the tumor necrosis factor alpha production of B cells was inhibited while the proportion of interleukin -10-producing regulatory B cell subsets was increased in mitoxantrone-treated patients [50]. Longitudinal data disclosed that the secretion of immunoglobulins was not altered [51]. In contrast, antibody-production was shown to be reduced in another study [50]. Moreover, it was observed that blood BAFF levels were elevated in mitoxantrone-treated MS patients [52].

*Azathioprine* itself is a prodrug, which active metabolites methyl-thioinosine monophosphate and thio-deoxyguanosine triphosphate inhibit purine synthesis in cells with a high turnover [53]. In doing so, total lymphocyte and B cell numbers are massively reduced upon treatment [54]. The duration of B lymphocyte depletion after intravenous infusion usually ranges from 3 to 8 months [55]. While transitional and memory B cells were found to be reduced efficiently during this time, the absolute number of plasmablasts remained unchanged [56,57,58]. In atopic dermatitis patients, azathioprine induced a significant decrease of serum interleukin-6 levels [59]. In mice, natural and antigen-specific IgM antibodies were dramatically suppressed in animals receiving azathioprine [60]. This was also shown for general immunoglobulin levels in patients with autoimmune hepatitis taking azathioprine [61]. Serum BAFF levels were reported to be higher in patients receiving azathioprine compared with patients receiving no immunosuppressive therapy [62].

Cyclophosphamide induces cell apoptosis, as its active metabolite phosphoramide mustard forms deoxyribonucleic acid crosslinks in cells with low levels of aldehyde dehydrogenases [63]. Differential expression of this enzyme explains why cyclophosphamide is immunosuppressive but not myeloablative [64]. In doing so, it reduces lymphocyte and B cell counts and also inhibits B cell activation, proliferation and differentiation after infusion [65]. Interestingly, neither counts of memory B cells nor counts of plasmablasts changed significantly after approximately 15 weeks of therapy with cyclophosphamide [66]. In patients receiving this treatment for longer periods of time, it does finally also reduce memory and plasmablast counts [67]. In lupus erythematosus, the contribution of cyclophosphamide to the development of hypogammaglobulinemia is being discussed [68]. Used in cancer chemoimmunotherapy, cyclophosphamide treatment increased BAFF in patients’ plasma [69].

### 2.2. Inhibition of Migration

Fingolimod is a sphingosine-1-phosphate receptor modulator, which is applied orally as a disease modifying treatment in multiple sclerosis [70]. By sequestering lymphocytes to peripheral lymphoid organs, this class of medications is thought to keep immune cells away from their sites of chronic inflammation. Several studies showed that it reduces the absolute lymphocyte count including B cells in a disproportionate manner [71,72,73]. While fingolimod decreases absolute memory B cell counts, it increases the proportion of transitional and regulatory B cells in MS patients [33,73,74,75,76,77]. Interestingly, the number of activated CXCR3^+^ CD138^+^ plasmablasts was significantly increased by fingolimod treatment [77]. Moreover, B cells of patients treated with fingolimod showed a higher production of the anti-inflammatory cytokines transforming growth factor beta and interleukin-10 [75]. In line, serum levels of interleukin-6 were found to be reduced [78]. Further, the expression of CXC chemokine receptor type 4 and CD80 on B cells decreased upon fingolimod treatment [73,74]. Compared to controls, IgG concentrations were lower in patients treated with fingolimod [79]. Moreover, fingolimod-treated MS patients had significantly higher serum concentrations of BAFF, which positively correlated with the proportion and the absolute number of transitional B cells in blood [80].

Siponimod, a direct successor of fingolimod for oral use with a similar mode of action was recently approved for the treatment of adults with secondary progressive MS with active disease evidenced by relapses or imaging features of inflammatory activity [81]. A recent study showed that siponimod reduces the total lymphocyte and B cell count, while increasing the proportion of regulatory B cells and decreasing the frequency of memory B cells [82]. In a murine myasthenia gravis model, no change was observed in either antibody titers or total or antigen-specific plasma cell populations after treatment [83]. In organotypic slice cultures, siponimod reduced levels of interleukin-6 and attenuated lysophosphatidylcholine-mediated demyelination [84]. Siponimod treatment had no relevant effect on IgG levels and antibody response after vaccination [85].

Ozanimod, the third-in-class sphingosine-1-phosphate receptor modulator, was approved for use in the United States in March 2020 and will soon be available in Europe as well [86,87]. Its effect on B cells yet needs to be examined.

Natalizumab is an intravenously-applicated humanized monoclonal antibody against the cell adhesion molecule α4-integrin [88]. Being infused in a 4-week interval, it is thought to reduce cell adhesion to endothelial cells and thereby prevent immune cells from crossing the blood–brain barrier [89]. In doing so, it was shown to massively increase lymphocyte and B cell counts in the peripheral blood of natalizumab-treated patients, with B cells counts being increased over-proportionally [90,91,92]. Interestingly, it enlarged the proportion of circulating immature, transitional, antigen-activated and memory B cells [33,92]. Perturbation of B cell homing in secondary lymphoid organs and direct effects on B cells are being discussed as reasons for this observation [90,93]. In this line, the expression of the activation markers CXC chemokine receptor type 4, CD25 and CD69 on B cells and the production of pro-inflammatory interleukin-6 and tumor necrosis factor were elevated in natalizumab-treated patients [90,91]. Thus, serum levels of interleukin-6 were found to be elevated in NMOSD patients treated with natalizumab when compared to patients treated with other therapies [94]. While the proportion of plasmablasts was decreased, their absolute number was found to be increased upon natalizumab treatment [90]. Regarding antibody production, natalizumab-treated MS patients showed lower levels of IgM and IgG and a decreased IgG-index compared to untreated controls [79,95,96,97]. Remarkably, natalizumab treatment did not alter serum BAFF levels in treated patients [98].

### 2.3. Depletion

Alemtuzumab, a monoclonal antibody against CD52, which is expressed on various immune cell types, was shown to reduce the total B cell count and proportion of B cells within the lymphocytic compartment [33,99]. Sequential intravenous infusions are given with a minimum interval of 12 month allowing a reconstitution of the depleted cells. The overshooting repopulation of B cells after depletion was demonstrated to be composed mainly of immature B cells, which are assumed to be enriched in B cells with a regulatory potential. Additionally, a marked and long-lasting depletion of memory B cells has been observed [100,101]. These aspects may explain how alemtuzumab controls the disease [101]. Additionally, the frequency and the absolute number of plasmablasts were significantly decreased compared to pretreatment levels [100]. In line with this, a recent study reported reductions of immunoglobulin levels (IgG, IgM and IgA) in serum and CSF following two courses of alemtuzumab in MS patients [102]. In alemtuzumab-treated renal transplant patients, serum BAFF levels were significantly above the normal range and remained elevated 2 years posttransplant in 78% of these patients. A BAFF-receptor on B cells was downregulated, suggesting ligand/receptor engagement [103]. Another study reported a rise in serum BAFF levels, which remained elevated for at least 12 months after alemtuzumab infusion [104].

Ocrelizumab and its precursor rituximab are capable of binding CD20 on B cells causing antibody-dependent cell-mediated cytotoxicity and, to a lesser extent, complement-dependent cytotoxicity [105]. Long-term B-cell-depleting therapy after ocrelizumab infusion with an interval of at least 5 months led to transiently reduced absolute lymphocyte counts, which recovered after the second cycle [106]. As a matter of fact, absolute B cell counts were diminished upon treatment [107,108]. Within the reappearing B cell pool, the frequency of interleukin-10 producing regulatory B cells was found to be increased, while the proportion of mature, memory and tumor necrosis factor-producing B cells was lower compared to untreated controls [4,107,108]. Due to their lack in CD20 expression, late plasmablasts and plasma cells are largely preserved [107]. However, decreases of immunoglobulin levels were seen in ocrelizumab-treated patients [109]. Effects of ocrelizumab on BAFF or interleukin-6 levels have not been published so far. As of rituximab, BAFF levels rose significantly during B-cell depletion in lupus erythematosus and rheumatoid arthritis patients and declined close to pretreatment levels upon B-cell repopulation [110,111]. Moreover, rituximab-mediated trogocytosis of B cells in vitro results in acute production and release of interleukin-6 along with increased levels of interleukin-6 mRNA [112]. Next, B cells returning after depletion secreted reduced levels of interleukin-6 in humans [113]. However, interleukin-6 serum levels were shown to be unaffected six months after B cell depletion [114].

### 2.4. Immunomodulation

Interferon-β was the first disease-modifying therapy to be approved in MS [115]. Depending on the subclass, it is applicated intramuscular or subcutaneously. In some cases, lymphopenia has been reported as a side effect [116,117]. However, it reduces the trafficking of inflammatory cells across the blood-brain barrier, probably by downregulating very late antigen-4 expression on lymphocytes [118]. This may explain the reported increase in the frequency of peripheral B lymphocytes upon treatment [119]. Interferon-β increases serum BAFF concentration [120,121]. Further, it increases the absolute number of regulatory, interleukin-10 producing transitional B cells and plasmablasts [119,122]. In contrast, the proportion of memory B cells and the expression if CD40 and CD80 on B cells were found to be decreased in interferon-β treated patients [122,123]. In MS, treatment with interferon-β was found to transiently increase serum interleukin-6 levels [124]. Additionally, a significant decrease in mean IgG concentrations has been reported [125]. Moreover, several studies demonstrated that interferon-β increases systemic BAFF levels in MS patients without increasing autoantibody production [121,126].

Dimethyl fumarate was recently shown to interfere with the aerobic glycolysis of activated lymphoid cells with a high metabolic turnover [127]. By this means, it effectively reduces absolute lymphocyte and B cell counts in treated patients [128,129,130]. Moreover, several studies confirmed that the orally given agent increases the proportion of transitional B cells while reducing memory and antigen-activated B cells with a disproportionate decrease of plasmablasts [128,131,132]. In this line, B cells of treated patients produce less interleukin-6 and tumor necrosis factor, resulting in a more anti-inflammatory cytokine profile and reduced levels of serum interleukin-6 [133,134]. In addition, the expression of molecules involved in antigen presentation (CD40, CD80 and CD86) was found to be decreased in patients treated with dimethyl fumarate [128,131]. Finally, serum levels of IgA, IgG2 and IgG3 were decreased upon treatment [135]. Studies report opposing effects of dimethyl fumarate on serum BAFF levels, which requires further investigations [131].

Glatiramer acetate is a subcutaneous-injected synthetic analogue of myelin basic protein, which is a component of the myelin sheath surrounding neurons [136]. While some studies reported no relevant modulation by glatiramer acetate regarding the overall lymphocyte and B cell count [90,98,137], others described an absolute and relative decrease of B cells in patients treated with glatiramer acetate [138,139]. Despite these discrepancies, several studies agree that glatiramer acetate treatment increases the proportion of interleukin-10 producing regulatory B cells and reduces the amount of memory B cells and plasmablasts [138,140,141]. Only one study reports that B cells of glatiramer acetate-treated patients produced higher amounts of interleukin-6 the longer they had been treated [138]. Furthermore, total IgG and IgM levels were shown to be elevated upon glatiramer acetate treatment [138], probably due to the observation that the majority of treated patients generate non-neutralizing anti-glatiramer acetate IgG antibodies [140]. While one study observed similar blood BAFF levels in patients with or without glatiramer acetate, two other reports that MS patients treated with glatiramer acetate had significantly higher serum BAFF levels [52,98,142].

## 3. MS Medications in NMOSD

As NMOSD has a progressive course already from the onset of the disease, it is vital to choose an appropriate therapy that efficiently prevents further relapses to avoid clinical exacerbations leading to lasting disability [143]. Until recently, there was no approved treatment for NMOSD [27], but steroids and immunosuppressive drugs known from MS therapy were used off-label. Unfortunately, several drugs used efficiently in MS are not effective or even harmful when prescribed to NMOSD patients. This section reviews trials, reports and case reports about the usage of MS medications in NMOSD.

### 3.1. Inhibition of Proliferation

Cladribine has not been reported to be effective or harmful in NMOSD. Currently, there is no data about the administration of cladribine in NMOSD.

Teriflunomide usage in NMOSD cannot be supported or discouraged as there are insufficient data until now [144]. Future studies will be needed to address this question.

Mitoxantrone is recommended as second-line treatment according to the Neuromyelitis Optica Study Group [144]. It was first studied in five patients with NMOSD, where expanded disability status scale (EDSS) scores improved, but two patients experienced a relapse [145]. In a more recent trial with 20 patients, the median pretreatment annualized relapse rate declined from 2.8 to 0.7 and the mean EDSS score declined from 5.6 to 4.4 with nausea being the most common adverse event [146]. Moreover, mitoxantrone was shown to dramatically decrease the frequency of relapses [147].

Azathioprine is a traditionally widely used medication in the therapy of NMOSD and was even recommended as first-line preventive treatment [148]. Studies showed a strong reduction of NMOSD relapse rate and reduced disability after years of treatment [149]. Another systematic review reported that the median EDSS decreased significantly from 7 to 6 on azathioprine [150]. However, discontinuation-rate after 18 months was almost 50%, suggesting very poor tolerability of this drug [151].

Cyclophosphamide has demonstrated beneficial effects in case reports in a limited number of NMOSD patients [152]. In a Japanese study, patients exhibited a median improvement in the EDSS score following cyclophosphamide treatment from 8.0 to 5.75. Adverse effects were observed in only one patient [153]. There is also a report about the successful use of cyclophosphamide in halting relapses in a patient with systemic lupus erythematosus-associated NMOSD [154]. However, there is a study that found no significant improvement in terms of EDSS score and reported that 80% of the treated patients had adverse effects on cyclophosphamide [150]. Thus, cyclophosphamide infusion is recommended as an alternative choice only for very difficult cases of NMOSD in Japan [155].

### 3.2. Inhibition of Migration

Fingolimod was shown to worsen NMOSD in some cases: In an NMOSD patient initially diagnosed as MS, multiple new lesions appeared in the brainstem and cerebral white matter after treatment was switched to fingolimod from interferon-β 1b. Following discontinuation of fingolimod, these lesions partly cleared, concomitantly with clinical improvement [156]. Another seropositive NMOSD patient was diagnosed of acute exacerbation of NMOSD ten days after treatment with fingolimod was initiated [157]. A third report is about a rebound after fingolimod cessation and a single daclizumab injection in a patient retrospectively diagnosed with NMOSD [158].

Siponimod treatment and outcomes for NMOSD have not been published so far, as the approval for MS was very recently. As fingolimod treatment was shown to coincide with acute exacerbation of the disease, one may assume that this is also the case for its successor siponimod (and ozanimod), which as a matter of course remains to be elucidated. [156,157].

Natalizumab was not able to control disease activity in NMOSD patients who were initially misdiagnosed as MS patients, when compared to patients treated with mitoxantrone or rituximab [94]. In the same line, five NMOSD patients who were initially misdiagnosed with MS and treated with natalizumab as escalation therapy after failure of first- or second-line immunomodulatory therapies for MS, displayed persisting disease activity and a total of nine relapses (median duration to relapse 120 days) after the start of treatment [159]. Another report described a patient with a relapsing optico-spinal demyelinating syndrome, negative for aquaporin-4 antibodies, who experienced a catastrophic brain relapse shortly after a single dose of natalizumab [160]. Further, massive astrocyte destruction was the neuropathological feature of a severe cerebral attack in a natalizumab-treated patient with relapsing myelitis and serum AQP4 antibodies [161].

### 3.3. Depletion

Alemtuzumab did not reach the expected effectiveness in NMOSD [15]. In a case report, a highly active antibody-positive NMOSD patient who was unstable on first-line treatment with rituximab did not benefit from alemtuzumab as an add-on therapy [162]. Some of the relapses were associated with the recurrence of B cells. Further, alemtuzumab itself may trigger antibody-driven diseases like Graves’ disease or idiopathic thrombocytopenic purpura [163]. In another report, an NMOSD patient treated with alemtuzumab experienced insidiously progressive nausea, vomiting, vision loss and marked extension of cortical, subcortical, and brainstem hyperintensities [164]. After she died about two years after the initial alemtuzumab infusion, acute, subacute and chronic demyelinating lesions were found at autopsy.

Ocrelizumab has been recently approved for the treatment of relapsing–remitting MS. To date, there is no published evidence regarding the treatment of NMOSD with ocrelizumab. Nonetheless, studies with its precursor rituximab, also being an anti-CD20 antibody, have shown profound beneficial effects on the annualized relapse rate and disability progression [165,166]. A recent meta-analysis with 577 patients suggested diminished mean annualized relapse rate ratio after rituximab therapy by 1.56 and a reduction in the mean EDSS score by −1.16 during rituximab treatment [167]. Moreover, rituximab showed no major safety issue and an acceptable tolerance [106]. These findings strongly suggest that ocrelizumab will also be effective in the treatment of NMOSD, which certainly needs to be proven.

### 3.4. Immunomodulation

Interferon-β treated NMOSD patients have been shown to experience clinical deterioration. In one study, 95% of patients exhibited an ineffective or exacerbated response to interferon-β treatment and the mean annualized relapse rates significantly increased after treatment [168]. Another report observed that giving interferon-β treatment to NMOSD patients may lead to a worsening of symptoms [169]. Moreover, antibodies against interferon-β were detected in 14 of 15 NMOSD patients and 6 of 15 patients developed neutralizing antibodies. However, progression of disease activity during treatment occurred irrespective of interferon-neutralizing antibody status [170]. It was also shown that the AQP4 antibody titers rose dramatically during treatment with interferon-β and then fell when immunosuppressive therapy with methotrexate and prednisolone was substituted [171]. Several other studies disclosed ineffectiveness or even worsening of symptoms of interferon-β-treated NMOSD patients [172,173,174,175].

Dimethyl fumarate is not commonly applied in NMOSD. However, a case report details a patient with NMOSD who was misdiagnosed with MS and experienced severe exacerbations 3 months following initiation of treatment with dimethyl fumarate [176]. There is another report of two cases of NMOSD who were misdiagnosed as MS and developed catastrophic relapses both 2 and 3 months after initiation of dimethyl fumarate [177]. Additionally, it was published that patients with NMOSD disorders were treated with dimethyl fumarate after de-escalation from rituximab. In patients with high disease activity, they had to return to rituximab within 6–9 months due to clinical ineffectiveness of dimethyl fumarate [25].

Glatiramer acetate is not beneficial for preventing attacks in most patients with NMOSD. In one study, the median EDSS increased and glatiramer acetate therapy was discontinued in 15 of 16 patients. Reasons were therapeutic inefficacy in 13 and post-injection skin reactions in two patients [178]. In another report, glatiramer acetate did not reduce the frequency of clinical attacks, while this was the case for azathioprine- and rituximab-treated NMOSD patients [179]. This placed into focus, glatiramer acetate may be considered clinically ineffective, but not deleterious in the treatment of NMOSD.

## 4. Ineffectiveness and Failure of MS Therapeutics in NMOSD

In conclusion, the following six drugs approved for MS are not effective or even harmful when used in NMOSD: Alemtuzumab, dimethyl fumarate, fingolimod, glatiramer acetate, interferon-β and natalizumab. In the following section, we attempted to compile drug-induced changes of B cell parameters and/or properties that presumably correlate to ineffectiveness in NMOSD therapy. In summary, we postulated the following distinctive features of MS therapeutics correlating to NMOSD treatment failure: elevation of the total B cell count, memory B cells, immunoglobulins, plasmablasts, serum interleukin-6 and serum BAFF.

### 4.1. Elevation of the Total B Cell Count

When comparing the effects of MS medications on the absolute B cell count, we observed that those drugs reducing or not altering B cells numbers in the blood, such as ocrelizumab, mitoxantrone and azathioprine were well tolerated by NMOSD patients (Table 1). However, compounds that induced an increase in the total B cell count, such as interferon-β, natalizumab and alemtuzumab, were shown to be ineffective or even harmful in NMOSD. This observation may be explained by recapitalizing the immuno-pathophysiology of NMOSD in comparison to MS: NMOSD is associated with astrocyte-targeting anti-AQP4 antibodies present in the patient’s blood [180]. Hence, an elevated number of blood B cells possibly contribute to NMOSD pathogenesis by providing an increased number of plasma cell progenitors, which subsequently foster a pathogenic humoral immune response. Moreover, antibody-independent B cell functions like antigen-presentation and provision of pro-inflammatory cytokines [28] may additionally quantitatively strengthen by the increased number of B cells contributing to disease progression. However, the question arises, why these B cell number-increasing medications are effective in MS. As shown above, natalizumab and interferon-β were revealed to reduce the trafficking of inflammatory cells across the blood–brain barrier by blocking or downregulating very late antigen-4 on immune cells respectively [181,182]. Thereby, these drugs are thought to downregulate inflammatory processes within the CNS in MS. In NMOSD pathogenesis, cellular infiltration plays a minor role compared to antibodies, whose production seems to be fostered by an increasing number of peripheral B cells. This may explain why an inhibition of migration is rather harmful in NMOSD. In the case of alemtuzumab, only B cells, not T cells, were shown to rapidly repopulate in the peripheral blood of treated patients [101]. Relapses in NMOSD, which is considered a peripheral B cell-driven disease, may therefore be more frequent than in alemtuzumab-treated MS patients.

### 4.2. Increase of Memory B Cells

Except for natalizumab, all approved MS medications proportionally decrease the frequency of activated or memory B cells, while most approved drugs were shown to increase the regulatory B cell subset. In doing so, they induce an-anti-inflammatory shift in the cytokine profile of B cells and promote a less activated B cell phenotype. Thus, memory B cells were shown to be major targets for effective immunotherapy in MS [33]. Natalizumab, which was shown to cause relapses in NMOSD, blocks the very late antigen-4 molecule on immune cells and thereby prevents CNS infiltration. As a possible explanation for the elevated memory B cell frequency, it was shown that natalizumab perturbs homing of activated B cells in secondary lymphoid organs [93]. Moreover, direct downstream effects within the B cell upon binding of natalizumab and indirect effects via directly activated T cells are being discussed [90]. Again, the peripheral activation of B cells leads to an increased number of plasma cells, which is associated with elevated secretion of pro-inflammatory cytokines, which potentially explains why natalizumab is not effective or even harmful in MS, but not in NMOSD. A recent study indicates that frequencies of peripheral B cells in NMOSD are shifted from regulatory to a more activated, memory B cell phenotype, which might plausibly explain why natalizumab is harmful in these diseases [183]. Moreover, memory B cells were shown to worsen the severity of NMOSD [184]. In rituximab-treated NMOSD patients, elevated CD27^+^ memory B cells were reported to be a reliable marker for biological and clinical relapses after depletion [185,186,187]. In contrast, a depletion of memory B cells was associated with a clinical response to rituximab treatment [188]. Interestingly, severe relapses after natalizumab cessation in MS are also associated with elevated memory B cell counts [90].

As a passing mention, long-term therapy with glucocorticoids was shown to elevate memory B cells after six months of IgG4-related disease [189]. As glucocorticoids are basically only used as pulsed therapy for relapse treatment in NMOSD, these potentially adverse effects may only be relevant in rare cases with prolonged glucocorticoid treatment [189].

### 4.3. Elevated Immunoglobulin Levels

In contrast to all other drugs approved for MS, glatiramer acetate was shown to increase IgG and IgM levels in the peripheral blood [138]. This may be due to the fact that the level of the B cell activating factor, which is thought to promote B cell survival and proliferation, is increased in the sera of glatiramer acetate-treated patients [142]. The fact that almost all patients treated with glatiramer acetate generate non-neutralizing anti-glatiramer acetate IgG antibodies may point towards the assumption that B cells are being triggered to differentiate into antibody-secreting plasma cells by this medication, possibly explaining its clinical ineffectiveness in NMOSD. As higher production of peripheral IgG does not automatically increase anti-AQP4-antibody levels, studies about the effect of MS medications on AQP4-antibodies are needed at this point.

### 4.4. Elevated Plasmablast Count

Plasmablasts are an intermediate stage in the differentiation of B cells to plasma cells, which, in contrast to fully differentiated plasma cells, are still capable of antigen presentation and cell proliferation, but produce lower amounts of antibodies [190]. As shown above, NMOSD-deteriorating interferon-β and natalizumab persistently increase the total plasmablast count and fingolimod elevates the proportion of activated CXCR3^+^ CD138^+^ plasmablasts. It has been reported that anti-AQP4 antibodies are mainly produced by plasmablasts, providing a possible mechanism by which these cells causally contribute to disease activity [18]. Additionally, the number of antibody-secreting peripheral plasmablasts was shown to correlate with the frequency of clinical relapses [191]. Intrathecally-produced antibodies derive from plasmablasts within the CNS, that were shown to be increased in the CSF after clinical relapses [19].

### 4.5. Elevated Serum Interleukin-6 Levels

As a proinflammatory cytokine, interleukin-6 is essential for the interaction between the initial and the acquired immune system. While interleukin-6 promotes B cell activation, B cells themselves are able to produce interleukin-6 to further foster inflammatory processes, especially by generating T helper 17 cells, which were shown to be main contributors to NMOSD pathology [192]. Serum levels of interleukin-6 were found to be increased by alemtuzumab shortly after infusion, interferon-β and natalizumab, which might explain their failure in NMOSD: on a cellular base, natalizumab specifically increases the interleukin-6 production of B cells, again foregrounding the pathological role of antibody-independent processes in NMOSD and the success of interleukin-6 receptor antagonists [90].

### 4.6. Elevated BAFF Levels in Non-Depleting Agents

High serum BAFF levels were shown to be associated with rising antibody levels in various autoimmune diseases by promoting B cell activation and differentiation [35]. When evaluating soluble BAFF levels, it must be considered that they inversely correlate with peripheral B cell numbers and the expression of BAFF receptors [34]. This may explain why B cell depleting agents (alemtuzumab, azathioprine, cyclophosphamide, mitoxantrone and rituximab) elevate BAFF levels. Causatively, two distinct mechanisms are being discussed [193]: one mechanistically related to the large decrease in receptors after B-cell depletion, and the other to delayed upregulation of the BAFF expression. Consequently, BAFF levels post depletion must be interpreted with caution and do not reflect the level of B cell activity in these patients. However, repeated rituximab infusions in systemic lupus erythematosus can result in a feedback loop characterized by ever-rising BAFF levels, surges in autoantibody production and worsening of disease [111]. Severe relapses upon rituximab treatment have been reported even for NMOSD patients [194]. In this line, elevated serum levels of interleukin-6 and BAFF were associated with relapses after rituximab treatment together with transient increases in anti-AQP4 titers [195].

In contrast, BAFF increases were also reported for immunomodulators with weaker effects on the total B cell count (fingolimod, glatiramer acetate and interferon-β). It is interesting to see that these medications are ineffective or even harmful in NMOSD. The fingolimod- and interferon-β-related increase of BAFF levels was shown to be not merely due to reduced consumption by BAFF receptors on B cells but involving active upregulation of transcription [98]. Elevated BAFF plasma levels were demonstrated to relatively increase transitional and decrease switched B cells as a result of the fact that BAFF receptor signaling aids the differentiation of immature B cells into transitional B cells [196]. It was claimed that BAFF may be a key factor of peripheral B cell immune response in NMOSD and reflect disease severity, possibly explaining the above-mentioned findings [197]. The amount of BAFF was shown to determine the size of the B cell pool and autoreactive B cells are especially dependent on BAFF [198]. Again, as NMOSD are considered peripheral autoimmune diseases, an increase of BAFF may causally explain relapses and treatment failure. In this regard, it must be noted that increased serum BAFF levels are indeed seen in patients with NMOSD [142]. In other T helper 17-driven autoimmune diseases like systemic lupus erythematosus, elevated BAFF levels were found to reflect disease activity [199]. BAFF was also shown to indirectly promote T helper 17 generation [200]. Elevated CSF levels of BAFF in NMOSD suggest additional intensified recruitment and activation of intrathecal B cells [183].

BAFF antagonists have been successfully introduced in the treatment of NMOSD-related diseases like systemic lupus erythematosus [201]. To our knowledge, there are no reports about BAFF antagonism in NMOSD, which might be an auspicious option. In MS, atacicept, a BAFF antagonist, has been proposed to be a promising treatment strategy based on animal models of MS, but caused unexpected increased inflammatory activity in MS patients, which might be related to a loss of interleukin-10-producting plasma cells [202,203,204,205]. Additionally, in contrast to NMOSD, BAFF levels of MS patients are not elevated and high serum BAFF levels reflect a more stable and effective MS treatment outcome, highlighting the above-mentioned pathophysiological differences of both diseases. [121].

### 4.7. Others

Firstly, apart from the effect on BAFF levels, clinical inefficacy of fingolimod may be due to the modulation of local sphingosine-1-phosphate receptors, which may lead to an increased permeability of the blood-brain barrier and enhance activation of astrocytes that have already been impaired [156]. Similar effects may be assumed for siponimod.

Secondly, patients with MS who developed autoimmunity upon alemtuzumab showed more than twofold higher serum interleukin-21 levels compared to patients with no autoimmunity [206]. A subsequent increase of interleukin-21 receptor-positive effector T cells may thereby increase the probability of encountering self-antigens and generating self-reactive T cells with a subsequent effect on B cells [163]. These mechanisms may also play a role in the failure of alemtuzumab in NMOSD.

Thirdly, it remains unclear, why severe relapses were seen in NMOSD during the first 3 months of dimethyl fumarate treatment [177]. None of the above-mentioned presumably NMOSD-worsening effects on B cells were seen in dimethyl fumarate-treated patients: B cell counts, memory B cells and immunoglobulin levels decreased upon treatment. Even in T helper 17-mediated experimental autoimmune encephalomyelitis animal model, resembling the human NMOSD model, dimethyl fumarate reduced clinical myelitis signs, demyelination and macrophage infiltration [207]. It has to be noted, that the described anti-inflammatory effects on B cells were measured at least 3 months after treatment initiation [129]. Additionally, beneficial effects of dimethyl fumarate on relapse rates were significant after 10 weeks and further amplified in the months after. Reduction of lesion activity was significant 12 weeks after dimethyl fumarate initiation [208,209]. Thus, the deterioration of NMOSD patients within the first 3 months of DMF treatment may rather be attributed to the initial lack of dimethyl fumarate efficacy combined with the aggressive nature of NMOSD pathophysiology.

## 5. Novel and Future NMOSD Treatments

Our findings support the assumption that NMOSD therapy is more efficacious and tolerable when it targets humoral responses of the immune system [162]. This goes along with the current understanding of differences in the pathophysiology of MS and NMOSD as presented in Figure 1. As reviewed above, azathioprine, mitoxantrone, cyclophosphamide and rituximab are currently used as off-label treatment options in NMOSD. Additionally, intravenous IgG have been proven to be effective in some antibody-mediated autoimmune diseases [210]. However, the evidence for their use in NMOSD is relatively little: in a study with eight patients, it was observed that the mean relapse rate and the EDSS declined significantly upon intravenous IgG treatment [211]. Interestingly, intravenous IgG were also shown to be effective in acute NMOSD relapses [212].

Treatment of relapses in NMOSD usually includes a pulsed intravenous methylprednisolone therapy for 3–5 days and can be extended in severe relapses [213]. For steroid-refractive patients with severe symptoms, therapeutic plasma exchanges are carried out every other day up to a total of seven times [214]. An initial combination of both options was associated with improved outcomes compared with delayed initiation of plasma exchange following glucocorticoid treatment [215]. Oral prednisone or prednisolone can help preventing disease progression before the preventive treatment works reliably.

The above-mentioned relatively broad treatment approaches commonly go along with intolerances and side effects. Thus, progress in the understanding of pathologic mechanisms in NMOSD has led to the development of novel therapeutic approaches, including the prevention of complement activation, depletion of antibody-producing cells and interference with interleukin-6 signaling [27]. Very promising results of phase 2 and 3 studies may fundamentally change NMOSD therapy in the near future. In 2019, eculizumab, which will be discussed below, was approved by the Food and Drug Administration and the European Medicines Agency in adult NMOSD patients who are AQP4 antibody positive, being the first approved treatment option [216]. Very recently, satralizumab has been approved by Japan’s Ministry of Health, Labor and Welfare for both adults and children with NMOSD [217]. In June 2020, inebilizumab was approved by the Food and Drug Agency [218]. The effects of these emerging novel treatment options in NMOSD on B cells are summarized in Table 1 but need to be examined in a greater detail in future studies.

### 5.1. Complement Factor C5 Antibodies

Eculizumab prevents the cleavage of the complement factor C5 into its subunits and thereby blocks the AQP4-antibody-triggered complement cascade. Thus, inflammation, the formation of the membrane attack complex along with subsequent astrocyte destruction and neuronal injury are suppressed by eculizumab [219,220]. Before its approval for NMOSD, this humanized monoclonal antibody had been approved for myasthenia gravis, paroxysmal nocturnal hemoglobinuria and atypical hemolytic uremic syndrome [221]. In a phase 3 trial including 143 AQP4-antibody positive NMOSD patients, eculizumab significantly suppressed the adjudicated annualized relapse rate, while there were no benefits on disability progression during the short period of the trial [222]. Eculizumab decreased serum interleukin-6 levels in patients with paroxysmal nocturnal hemoglobinuria and increased B cell counts [223,224]. Effects on the B cell compartment in NMOSD patients are not yet assessed in detail and may strongly differ from those in paroxysmal nocturnal hemoglobinuria due to a completely different pathophysiology.

Ravulizumab, another complement factor C5 inhibitor, which is approved for the treatment of paroxysmal nocturnal hemoglobinuria, is also considered a therapeutic option for NMOSD [225]. There is an ongoing phase-III study to evaluate the efficacy and safety of ravulizumab for the treatment of adult participants with NMOSD (NCT04201262). To date, the effect of C5-inhibitors on peripheral B cells remains unclear.

### 5.2. B Cell-Depleting Antibodies

Inebilizumab is a monoclonal antibody against CD19, leading to a broad depletion of B cells including plasmablasts and plasma cells, which are thought to produce autoantibodies in NMOSD [226]. Remarkably, inebilizumab reduced the relapse rate by 73% in both seropositive and seronegative NMOSD patients and reduced disability worsening by 15.5% in a recent phase 2/3 trial along with a favorable safety and tolerability profile [218].

Ofatumumab is an entirely human anti-CD20 antibody, which targets an epitope completely distinct from that of rituximab, leading to a dose-dependent depletion of B cells [227]. There is a promising report about a marked reduction in the relapse rate in a single patient [194].

### 5.3. Interleukin-6 Receptor Antibodies

Toclizumab and satralizumab are monoclonal antibodies against the interleukin-6 receptor. As reviewed above, this pro-inflammatory cytokine is produced by a large number of cells, including B and T cells, monocytes and fibroblasts [228]. Interleukin-6 has been found to be elevated in the CSF and serum of NMOSD patients and is thought to induce T cell activation, generation of autoreactive T helper 17 cells and AQP4-antibody secretion by plasmablasts [229,230]. In a first pilot study with seven patients, the annualized relapse rate decreased from 2.9 to 0.4 while EDSS, neuropathic pain, and general fatigue also declined significantly [231]. Similar effects were shown in a report, where toclizumab significantly reduced the annualized relapse rate from 4.0 to 0.4 and the median EDSS from 7.3 to 5.5 [232]. In a more recent trial with 118 included patients, 92% of treated NMOSD patients were relapse-free after 48 weeks compared to 69% in the azathioprine group with similar findings regarding disability progression [233]. Neither did treatment alter lymphocyte, B cell, transitional B cell and plasmablast count, nor did it alter BAFF levels of treated patients [234,235]. However, Toclizumab was shown to reduce memory B cell subsets in rheumatoid arthritis patients as well as IgA and IgG serum levels [236]. Interestingly, it was found that toclizumab treatment does not change interleukin-6 serum levels in rheumatoid arthritis patients [237]. Satralizumab was designed to improve the pharmacokinetics of toclizumab using antibody-recycling technology [238]. In two phase 3 studies, satralizumab significantly reduced the risk of protocol-defined relapses, but had significantly less effect on seronegative patients, which requires further analysis [233]. Its effect on B cells may be similar to those of toclizumab, but yet need to be examined.

### 5.4. Others

Aquaporumab competes with AQP4-antibodies for AQP4 binding. Due to a mutated Fc portion, it does not activate complement and cell-dependent mediated cytotoxicity [239]. In a preclinical study, aquaporumab was already proven to efficiently compete with pathological AQP4-antibodies [240]. Very recently, high affinity aquaporumab, which was generated by affinity maturation using saturation mutagenesis, was shown to block cellular injury caused by NMO patient sera in AQP4-expressing cell cultures [241].

Sivelestat, an inhibitor of neutrophil elastase, reduced lesion formation in animal models of NMOSD [242]. Dominant presence of neutrophils in inflammatory infiltrates in NMOSD and elevated neutrophil counts in the CSF may explain these findings [243,244].

Cetirizine, a selective inhibitor of the histamine receptor 1, significantly reduced eosinophil infiltrates and lesion formation in a murine NMOSD model [245]. It was found that eosinophil infiltration is a prominent feature of NMOSD lesions and eosinophils were shown to be elevated in the CSF of NMOSD patients [246]. When cetirizine was given as add-on therapy to standard treatment of NMOSD treatment, the annualized relapse rate was reduced fourfold after one year [247].

Bortezomib treatment in NMOSD was tried with the intent to enhance apoptotic death of AQP4-antibody-producting plasma cells. As a selective inhibitor of the 26S proteasome subunit, bortezomib stabilized or improved disease course in 4/5 otherwise refractory patients and diminished plasma cells as well as serum anti-AQP4-antibody levels [248].

Autologous stem cell transplantation is occasionally used in severe cases of NMOSD. Currently available data suggest that this procedure can reduce inflammatory activity in the short term, but a clear majority of the patients will relapse within 5 years [249]. There are also murine data showing that autologous hematopoietic stem cell transplantation is likely ineffective in NMOSD [250].

## 6. Conclusions

Since most approved MS medications were not primarily designed to target B cells, but T cells, their impact on B cells was long time disregarded. However, recent research demonstrates that due to the different modes of action, effects on B cells induced by these drugs vary strongly; they range from neutralization of B cell functions to activation of pro-inflammatory responses. In MS, CNS-infiltrating immune cells induce a potentially reversible oligodendrocytopathy with subsequent axonal damage, suggesting an infiltration-driven pathophysiology. In contrast, in seropositive NMOSD, peripherally produced antibodies against aquaporin-4 on astrocytes are the disease-drivers, while cellular infiltration plays a minor role. Having these differences in mind, it is not surprising that the therapeutic efficacy of B cell-modulating drugs differs between MS and NMOSD. Hence, when NMOSD patients were deliberately or unintendedly treated with MS medications, some of these therapeutic approaches resulted in ineffectiveness or clinical deterioration. In an attempt to correlate clinical ineffectiveness in AQP4 positive NMOSD to changes in the B cell compartment, we found the following B cell-related parameters associated with treatment failure in NMOSD:Increase of the total B cell count;Elevated proportion of memory B cells;Elevated proportion of plasmablasts;Increased immunoglobulin production;Elevated serum interleukin-6 levels;Elevated serum BAFF levels.

We postulated that B cell-related parameters should be considered important factors when it comes to the development or approval extension of potential NMOSD medications. Of note, the pharmacological effect on other immune cells and their interaction with potentially altered B cells also determine the clinical outcome of the respective medications and should be taken into account. Furthermore, it needs to be determined how a combination or a single effect of one of the above-listed parameters interferes with NMOSD pathophysiology. Interestingly, satralizumab, being one of the now approved NMOSD medications, does indeed target the above-mentioned interleukin-6 and points towards an essential, promising change of NMOSD therapy in the near future.

## Figures and Tables

**Figure 1 ijms-21-05021-f001:**
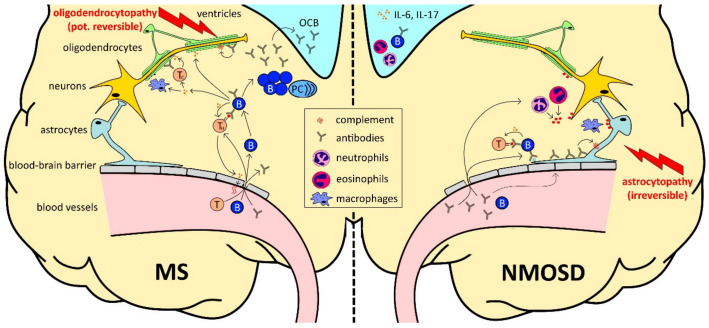
The role of B cells (B) in multiple sclerosis (MS) and neuromyelitis optica spectrum disorders (NMOSD). B cell development begins in the bone marrow. Transitional B cells enter the circulation and mature into naïve B cells after final maturation and selection in the spleen. In secondary lymphoid organs, upon antigen recognition, they differentiate into plasma cells (PC) or memory B cells with the help of T cells (T). (1) In MS (infiltration-driven; left side), B cells themselves cross the disrupted blood–brain barrier and form plasma cell clones within the brain parenchyma and meningeal tertial B cell follicles, which resemble the local source of oligoclonal bands (OCB), contributing to inflammation and destruction of the myelin sheet in the central nervous system. Infiltrating B cells also serve as highly effective antigen-presenting cells, leading to optimal antigen-specific T cell, cytokine production and further disruption of the blood–brain barrier. (2) In NMOSD (antibody-driven; right side), peripheral plasma cells are the main source of high amounts of aquaporin 4-specific antibodies, targeting astrocytic end feet after crossing the disrupted blood–brain barrier, followed by an activation of the complement cascade. As a consequence of astrocyte destruction, cortical demyelination occurs at late stages of the disease. Interleukin-6 (IL-6), which was found to be increased in NMOSD, promotes the generation of T helper 17 cell lineage, secreting interleukin-17 (IL-17).

**Table 1 ijms-21-05021-t001:** Effect of approved disease-modifying multiple sclerosis medications on B cell-related parameters in humans.

Agent	Lymphocyte Count	B Cell Count	% Transitional/Regulatory BC	% Memory/Activated BC	Plasmablast Count	Interleukin-6 (Serum)	IgG (Serum)	BAFF (Serum)
**MS: Inhibition of Proliferation**								
Cladribine	↓	↓	↑	↓	⇿	⇿	↓	–
Teriflunomide	↓	↓	⇿	⇿	↓	–	↓	–
Mitoxanthrone	↓	↓	↑	↓	–	–	↓	↑
(Azathioprine)	↓	↓	↓	↓	⇿	↓	↓	↑
(Cyclophosphamide)	↓	↓	–	↓	↓	–	↓	↑
**Inhibition of Migration**								
Fingolimod	↓	↓	↑	↓	↑^1^	↓	↓	↑
Siponimod	↓	↓	↑	↓	⇿^2^	↓^3^	⇿	–
Natalizumab	↑	↑	↑	↑	↑	↑	↓	⇿
**Depletion**								
Alemtuzumab	↓	↑^4^	↑	↓	↓	↑^5^	↓	↑
Ocrelizumab	↓	↓	↑	↓	⇿	⇿^6^	↓	↑^6^
**Immunomodulation**								
Interferon-β	⇿	↑	↑	↓	↑	↑	↓	↑
Dimethyl fumarate	↓	↓	↑	↓	↓	↓	↓	⇿
Glatiramer acetate	⇿	⇿	↑	↓	↓	↑^7^	↑	↑
**NMOSD: B Cell Depletion**								
Rituximab	↓	↓	↑	↓	⇿	⇿	↓	↑
Inebilizumab	↓	↓	–	–	↓	–	↓	–
Ofatumumab	↓	↓	–	–	–	–	–	–
**Interleukin-6 Receptor Antibodies**								
Tocilizumab	⇿^8^	⇿	⇿	↓^8^	⇿	⇿^8^	↓	⇿^8^
Satralizumab	–	–	–	–	–	–	–	–
**Complement Factor C5 Antibodies**								
Eculizumab	↑^9^	↑^9^	–	–	–	↓^9^	–	–
Ravulizumab	–	–	–	–	–	–	–	–

^1^ Increase of activated plasmablasts; ^2^ in a murine myasthenia gravis model; ^3^ in an organotypic slice culture; ^4^ hyper-repopulation after initial depletion; ^5^ after long-term treatment; ^6^ in rituximab-treated patients; ^7^ by B cells after long-term treatment; ^8^ in rheumatoid arthritis patients; ^9^ in paroxysmal nocturnal hemoglobinuria patients; IgG = immunoglobulin G; BC = B cells; BAFF = B cell activating factor; MS = multiple sclerosis; NMOSD = neuromyelitis optica spectrum diseases; ↑ = increased; ⇿ = unchanged; ↓ = reduced.

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
