# Peer review of "Differential Effects of MS Therapeutics on B Cells—Implications for Their Use and Failure in AQP4-Positive NMOSD Patients"

_ijms, 2020, doi:10.3390/ijms21145021_

Round 1

Reviewer 1 Report

This is an interesting review on the role of B cells in NMOSD and MS. This article highlights the effect of current and future therapies on B cells and their associated activating factors.

I have no particular concern with this work which is clearly exposed and well detailed.

I would suggest clarifying that the population of NMOSD patients concerned by this work is AQP4 + and not MOG+ or double seronegative. I would change the title accordingly.

I have doubts about the deleterious effect of GA on NMOSD. To the best of my knowledge, studies have shown no effect but no worsening of the inflammatory process.

Finally, I would suggest adding to Table 1 eculizumab, satralizumab, tocilizumab, inebilizumab, ravulizumab as well as the other molecules acting on B cells such as ofatumumab and rituximab, which are already used to treat patients with NMOSD.

I would add a short paragraph on attack treatments, including methylprednisolone and plasmapheresis.

Author Response

I would suggest clarifying that the population of NMOSD patients concerned by this work is AQP4 + and not MOG+ or double seronegative. I would change the title accordingly.

Thanks for this constructive feedback. In our introduction, we explain that MOG+ encephalomyelitis is now considered a seperate disease entity, immuno-pathogenetically distinct from both MS and NMOSD (lines 82-87). In our revised version, we changed the title of the review to "Differential effects of MS therapeutics on B cells - implications for their use and failure in AQP4-positive NMOSD patients" (lines 2-4) to clarify the serostatus of the population concerned by this work. Accordingly, we edited parts of the abstract and the conclusion:

"effective and safe in aquaporin-4 antibody positive NMOSD(line 18)

"seropositive NMOSD" (line 22)

"In seropositive patients, which represent 70% of all NMOSD patients" (line 43)

"their use or failure in AQP4-positive NMOSD" (line 99)

"in seropositive NMOSD, peripherally produced antibodies" (lines 653-654)

"AQP4 positive NMOSD" (line 659)

I have doubts about the deleterious effect of GA on NMOSD. To the best of my knowledge, studies have shown no effect but no worsening of the inflammatory process

We appreciate this comment, as GA was indeed not shown to have deleterious effects in NMOSD, while it clearly was uneffective when tested in NMOSD patients. To emphasize this in the review, we added the following paragraph to the GA subsection:

"This placed into focus, glatiramer acetate may be considered clinically ineffective, but not deleterious in the treatment of NMOSD." (lines 405-407).

In the immunoglobulin passage, we added to the text about GA the following:

"possibly explaining its clinical ineffectiveness in NMOSD" (lines 468-469). 

Finally, I would suggest adding to Table 1 eculizumab, satralizumab, tocilizumab, inebilizumab, ravulizumab as well as the other molecules acting on B cells such as ofatumumab and rituximab, which are already used to treat patients with NMOSD.

Thank you for this constuctive idea. We think that adding NMOSD drugs to the summarizing table will help the reader connecting the effects on B cells with new treatment approaches in NMOSD on the one hand and the patophysiological differences of MS and NMOSD on the other hand. Thus, we amended Eculizumab, Ravulizumab, Inebilizumab, Ofatumumab, Rituximab, Satralizumab and Toclizumab to Table 1, grouping them by their mechanism of action.

Also, effects of Eculizumab and Toclizumab on B cells were added:

"Eculizumab decreased serum interleukin-6 levels in patients with paroxysmal nocturnal haemoglobinuria and increased B cell counts [224,225]. Effects on the B cell compartment in NMOSD patients are not yet assessed in detail and may strongly differ from those in paroxysmal nocturnal haemoglobinuria due to a completely different pathophysiology." (lines 588-591)

"Neither did treatment alter lymphocyte, B cell, transitional B cell and plasmablast count, nor did it alter BAFF levels of treated patients [235,236]. However, Toclizumab was shown to reduce memory B cell subsets in rheumatoid arthritis patients as well as IgA and IgG serum levels [237]. Interestingly, it was found that toclizumab treatment does not change interleukin-6 serum levels in rheumatoid arthritis patients [238]. (lines 615-619) 

In addition, we inserted paragraphs about Ravilizumab and Ofantumumab to round up this section:

"Ravulizumab, another complement factor C5 inhibitor, which is approved for the treatment of paroxysmal nocturnal haemoglobinuria, is also considered a therapeutic option for NMOSD [226]. There is an ongoing phase-III study to evaluate the efficacy and safety of ravulizumab for the treatment of adult participants with NMOSD (NCT04201262). To date, the effect of C5-inhibitors on peripheral B cells remains unclear." (lines 592-596)

"Ofatumumab is an entirely human anti-CD20 antibody which targets an epitope completely distinct from that of rituximab, leading to a dose-dependent depletion of B cells [228]. There is a promising report about a marked reduction in relapse rate in a single patient [195]." (lines 602-604).

I would add a short paragraph on attack treatments, including methylprednisolone and plasmapheresis.

This is a very good suggestion, as it gives the reader a broader picture of NMOSD therapy. Thanks for this idea. In the NMOSD therapy section, we included a short paragraph about relapse treatment in NMOSD:

"Treatment of relapses in NMOSD usually includes a pulsed intravenous methylprednisolone therapy for 3-5 days and can be extended in severe relapses [214]. For steroid-refractive patients with severe symptoms, therapeutic plasma exchanges are carried out every other day up to a total of seven times [215]. An initial combination of both options was associated with improved outcomes compared with delayed initiation of plasma exchange following glucocorticoid treatment [216]. Oral prednisone or prednisolone can help preventing disease progression before the preventive treatment works reliably." (lines 561-567)

Reviewer 2 Report

The authors comprehensively described the B cell involvement in the pathophysiology, and the mechanism of action of each therapeutic agent for MS and NMOSD.

This is a very interesting and comprehensive review. There are some points the authors should consider.

  1. Line 304 (p7)

Several monoclonal antibodies were already approved for NMOSD. Eculizumab was approved in the US and EU in 2019. Satralizumab has been approved in Japan and inebilizumab was approved in the US in June 2020, respectively. This update will be informative for readers.

  1. Line 580 (p13)

As far as I know, AQP4-antibody secretion by plasmablast was firstly reported in another paper (Chihara N, et al. Proc Natl Acad Sci U S A 2011;108:3701–3706). Moreover, a first pilot study of tocilizumab for NMOSD was reported by Japanese group (Araki M, et al. Neurology 2014; 82:1302–1306). These articles should be cited in this review.

Author Response

Several monoclonal antibodies were already approved for NMOSD. Eculizumab was approved in the US and EU in 2019. Satralizumab has been approved in Japan and inebilizumab was approved in the US in June 2020, respectively. This update will be informative for readers.

Thanks for this comment. We added the following paragraph to the manuscript to keep the interested reader informed about the latest developments in the rapidly evolving field of NMOSD therapy:

"In 2019, eculizumab, which will be discussed below, was approved by the Food and Drug Administration and the European Medicines Agency in adult NMOSD patients who are AQP4 antibody positive, being the first approved treatment option [217]. Very recently, satralizumab has been approved by Japan’s Ministry of Health, Labour and Welfare for both adults and children with NMOSD [218]. In June 2020, inebilizumab was approved by the Food and Drug Agency [219]. The effects of these emerging novel treatment options in NMOSD on B cells are summarized in Table 1 but need to be examined in a greater detail in future studies." (lines 573-579)

As far as I know, AQP4-antibody secretion by plasmablast was firstly reported in another paper (Chihara N, et al. Proc Natl Acad Sci U S A 2011;108:3701–3706). Moreover, a first pilot study of tocilizumab for NMOSD was reported by Japanese group (Araki M, et al. Neurology 2014; 82:1302–1306). These articles should be cited in this review.

We appreciate this constructive feedback and added these relevant references to the manuscript:

"It has been reported that anti-AQP4 antibodies are mainly produced by plasmablasts, providing a possible mechanism by which these cells causally contribute to disease activity [18]." (lines 476-478) 

"In a first pilot study with seven patients, the annualized relapse rate decreased from 2.9 to 0.4 while EDSS, neuropathic pain, and general fatigue also declined significantly [231]." (lines 609-611)